# Particles Dynamics in Schwarzschild like Black Hole with Time Contracting Horizon

Muhammad Atif Khan [1], Farhad Ali [1,*], Nahid Fatima [2] and Mohamed Abd El-Moneam [3]

[1] Institute of Numerical Sciences, Kohat University of Science and Technology, Kohat 26000, Pakistan
[2] Department of Mathematics and Sciences, Prince Sultan University, P.O. Box 66833, Riyadh 11586, Saudi Arabia
[3] Department of Mathematics, Faculty of Science, Jazan University, Jazan 45142, Saudi Arabia
* Correspondence: farhadali@kust.edu.pk

**Abstract:** The black holes radiate their mass and energy in the form of gravitational waves and Hawking-radiation, which lead to a decrease in the mass and energy of the black holes. During the formation of gravitational waves and Hawking radiation, the mass and energy of black holes reduce continuously with the passage of time $t$. For this reason the metric tensor of the black hole should depends on time $t$. In this work, a time-dependent term is introduced in the horizon of black hole without losing its symmetry structure by using the approximate Noether symmetry equation. The time-dependent term affects the effective potential, effective force, and all the dynamic features of the black hole. They are discussed for neutral and charged particles. Profiles of the escape velocity of colliding particles are also taken into consideration. Lyapunov exponent is used to check the stability of the orbits of the black hole. Hawking temperature, Bekenstein entropy, Komar energy, and specific energy at horizon of the black hole are discussed in this scenario.

**Keywords:** thermodynamics; evaporating black holes; approximate Noether symmetries; energy and momentum conservations

**MSC:** 83C15; 83C25; 83C35; 83C40; 83C57





## 1. Introduction

There are numerous predictions of the general theory of relativity. One of these predictions is the formation of the gravitational waves. Einstein and his colleagues worked on the existence of the gravitational waves [1–3]. The LIGO-VIRGO collaboration already announce its existence [4,5]. Heavy and accelerated celestial bodies, such as black holes and neutron stars, produce these types of waves. Numerous researchers are working on the gravitational waves, some of which are given in [6–10]. According to the theory of general relativity, the celestial bodies form their own spacetimes in which they produce certain allowable paths for other comparatively small bodies to move.

The connections of thermodynamics and black holes are quite interesting and surprising. As the Hawking radiations identifies black holes as thermodynamical objects associated with temperature, entropy, surface area, and surface gravity at event horizons. Hawking, Bardeen, and Bekenstein [11,12] made some major and pioneering contributions to the thermodynamics of black holes. The theory of general relativity explains that black holes emit black body radiation known as Hawking radiation. Black holes evaporate in the form of gravitational waves and Hawking radiation which reduces the internal mass energy of the black hole. These observations lead to the fact that black holes, not only accreting mass from its surroundings, but also emits its mass energy as gravitational waves and Hawking radiation [13]. Due to Hawking radiation, a black hole evaporates, but this evaporation will take billions of years. Such black holes would be detectable from our Earth only if it blows up within the solar system. Spherically symmetric solutions of the

Einstein gravitational field equations are important because the black hole solutions of these equations are mostly spherical spacetime [14,15]. These objects (black holes) represent the regions of spacetime that are invisible from their exterior. The first such known exact solutions to Einstein field equations was found in 1916 by a well-known German physicist and astronomer Karl Schwarzschild.

The studies of time-like and null circular geodesics in black holes convey key information on the background geometry. Null geodesics is the curved path that photons follow. The optical appearance of a star going through the gravitational collapse is totally dependent on the circular unstable null geodesic. Null geodesics play a vital role in the explanation of characteristic modes of a black hole [16,17]. Null geodesics are also very useful to explain the characteristic modes of a black hole, such as quasinormal modes (QNM), which are the modes of energy dissipation of a perturbed object/field [18]. Orbital time period of time-circular geodesics is higher than the null-circular geodesics, so the null circular geodesics provides fastest track to circle of a black hole [19]. Stability and instability of circular orbits (for time-like and light-like/null geodesics) at equatorial plane can be traced by Layapunov exponent [20,21]. In addition to this, another type of circular orbits is known as homoclinic orbits, lying between dynamically unstable and stable orbits which are closed to circular orbits and lies in the intersection of the stable manifold and the unstable manifolds [22,23]. Another type of orbits, i.e., chaotic orbits are discovered for fast spinning black holes [24] and innermost circular orbits (ISCO) [25] also identify the origin of dynamical instability.

The perturbation methods are used for the solutions of many problems [26]. In this article, the horizon of Schwarzschild black hole is perturbed by time dependent term. The study explores the geodesics of evaporating a Schwarzschild-like black hole. We perturbed the metric by a general time-dependent term without losing the symmetry structure of the black hole by using the famous Noether symmetry equation. The paper is arranged in the following order.

Section 1 contains the literature servery which provides ground to this study. In Section 2, the perturbed Schwarzschild metric, along with its Lagrangian, the corresponding Noether symmetries, and conservations laws, is given. In Section 3, we discussed the effective potential, effective force, escape velocity of colliding particles, and stability of inner most circular orbits (ISCO) at equatorial plane. The dynamics of charged particles is given in Section 4. In Section 5, a discussion of the Hawking temperature, Bekenstein–Hawking entropy, Komar energy, and surface gravity is given. The pressure at the perturbed horizon of evaporating Schwarzschild black hole is given in same section. Conclusions and discussions are given in Section 6.

## 2. Time Contracting Schwarzschild Black Hole and Its Approximate Noether Symmetries

The background metric of Schwarzschild black hole with is defined as:

$$ds^2 = \mathcal{H}(r)dt^2 - \frac{1}{\mathcal{H}(r)}dr^2 - r^2 d\Theta^2, \tag{1}$$

with

$$\mathcal{H}(r) = 1 - \frac{2M}{r}, \qquad d\Theta^2 = d\theta^2 + \sin^2\theta \, d\phi^2.$$

Here, $M$ represents mass of the black hole. Coordinates $t$ and $r$ refers to time and radial coordinates, with polar and azimuthal angles $\theta$ and $\phi$, respectively. The Lagrangian corresponds to Equation (1) is

$$\pounds = \mathcal{H}(r)\dot{t}^2 - \frac{1}{\mathcal{H}(r)}\dot{r}^2 - r^2(\dot{\theta}^2 + \sin^2\theta\dot{\phi}^2). \tag{2}$$

In Equation (2), the symbol "·" denotes derivative with respect to proper distance $s$. Since due to gravitational waves and Hawking-radiation a black hole loses its mass and energy which is known as black hole evaporation phenomena. Therefore, the metric of the black hole should depends on time $t$. For this purpose we perturb the black hole by introducing time inside the horizon as

$$ds^2 = \mathcal{H}(r,t)dt^2 - \frac{1}{\mathcal{H}(r,t)}dr^2 - r^2 d\Theta^2. \tag{3}$$

where the function $\mathcal{H}(r,t)$ is defined as

$$\mathcal{H}(r,t) = 1 - \frac{2Mf(t)}{r}. \tag{4}$$

Lagrangian corresponding to Equation (3) takes the form

$$£ = \mathcal{H}(r,t)\dot{t}^2 - \frac{1}{\mathcal{H}(r,t)}\dot{r}^2 - r^2(\dot{\theta}^2 + \sin^2\theta\dot{\phi}^2). \tag{5}$$

Using the Lagrangian given in Equation (10) in the Noether symmetry equation

$$\mathbf{X}^{[1]}£ + D\xi(£) = DA, \tag{6}$$

where $D$ is the total differential operator and $A$ is gauge function, we have the system of 19 partial differential equations

$A_s = 0,\ \xi_t = \xi_r = \xi_\theta = \xi_\phi = 0,\ 2\eta^1 + 2\eta_\theta^2 - \xi_s r = 0,$

$2\eta_s^0\left(1 - \frac{2Mf(t)}{r}\right) - A_t = 0,\ 2\eta_s^t\left(1 - \frac{2Mf(t)}{r}\right)^{-1} + A_r = 0,$

$2\eta_s^2 r^2 + A_\theta = 0,\ 2\eta_s^3 r^2 \sin^2\theta + A_\phi = 0,\ \eta_\theta^3 \sin^2\theta + \eta^2 - \phi = 0,$

$\eta_\theta^1\left(1 - \frac{2Mf(t)}{r}\right)^{-1} + \eta_r^2 r^2 = 0,\ \eta_\phi^1\left(1 - \frac{2Mf(t)}{r}\right)^{-1} + \eta_r^3 r^2 \sin^2\theta = 0,$

$\eta_\phi^0\left(1 - \frac{2Mf(t)}{r}\right) - \eta_t^3 r^2 \sin^2\theta = 0,\ \eta_r^0\left(1 - \frac{2Mf(t)}{r}\right) - \eta_t^1\left(1 - \frac{2Mf(t)}{r}\right)^{-1} = 0, \tag{7}$

$\eta_\theta^0\left(1 - \frac{2Mf(t)}{r}\right) - \eta_t^2 r^2 = 0,\ 2\eta^1 + 2\eta^2 r \tan\theta + 2\eta_\phi^3 r - \xi_s = 0,$

$-2\frac{\eta^0 f_t(t)}{r}\left(1 - \frac{2Mf(t)}{r}\right)^{-1} + 2\frac{\eta^1 f(t)}{r^2}\left(1 - \frac{2Mf(t)}{r}\right)^{-1} - 2\eta_r^1 + \xi_s = 0,$

$-2\frac{\eta^0 Mf_t(t)}{r} + 2\frac{\eta^1 Mf(t)}{r^2} + 2\eta_t^0\left(1 - \frac{2Mf(t)}{r}\right) - \xi_s\left(1 - \frac{2Mf(t)}{r}\right) = 0.$

The solution of the system given in Equation (7) is

$$A = C_3,\ \xi = -\frac{C_1 s\gamma}{\tau} + C_2,\ \eta^0 = C_1\left(-\frac{\gamma t}{2\tau} + \frac{1}{2}\right),\ \eta^1 = -\frac{C_1 r\gamma}{2\tau},\ f(t) = 1 - \frac{\gamma t}{\tau},$$

$$\eta^2 = -C_5\cos(\phi) + C_6\sin(\phi),\ \eta^3 = C_4 + \cot(\theta)(C_5\sin(\phi) + C_6\cos(\phi)).$$

both time $t$ and characteristic time $\tau$ have equal dimensions which makes $\frac{t}{\tau}$ dimensionless and will reproduced the metric of Schwarzschild black hole and its Lagrangian as:

$$ds^2 = \left(1 - \frac{2M(1 - \frac{\gamma t}{\tau})}{r}\right)dt^2 - \left(1 - \frac{2M(1 - \frac{\gamma t}{\tau})}{r}\right)^{-1}dr^2 - r^2 d\Theta^2$$

$$£ = \left(1 - \frac{2M(1 - \frac{\gamma t}{\tau})}{r}\right)\dot{t}^2 - \left(1 - \frac{2M(1 - \frac{\gamma t}{\tau})}{r}\right)^{-1}\dot{r}^2 - r^2(\dot{\theta}^2 + \sin^2\theta\dot{\phi}^2) \tag{8}$$

The mass of the black hole in our case becomes

$$M = \mathcal{M}(1 - \frac{\gamma t}{\tau}). \tag{9}$$

The approximate Noether symmetry generators [27] corresponding to the Lagrangian given in Equation (8) take the form:

$$\mathbf{Y}_1 = \frac{1}{2}\frac{\partial}{\partial t} - \frac{\gamma}{\tau}\left(s\frac{\partial}{\partial s} + \frac{t}{2}\frac{\partial}{\partial t} + \frac{r}{2}\frac{\partial}{\partial r}\right), \quad \mathbf{Y}_2 = \frac{\partial}{\partial s}, \quad \mathbf{Y}_3 = \frac{\partial}{\partial \phi},$$

$$\mathbf{Y}_4 = \cos\phi\frac{\partial}{\partial \theta} - \cot\theta\sin\phi\frac{\partial}{\partial \phi}, \quad \mathbf{Y}_5 = \sin\phi\frac{\partial}{\partial \theta} + \cot\theta\cos\phi\frac{\partial}{\partial \phi}. \tag{10}$$

The conservation laws associated to the set of Noether symmetries in Equation (10) are:

### 3. Circular Null Geodesic at Equatorial Plane

From the Table 1 we have:

$$\dot{t} = \left[\frac{(1 + \frac{\gamma t}{\tau})}{\mathcal{H}(r,t)}\left(\mathcal{E} - \frac{\gamma t}{\tau}(s\mathcal{L} + \frac{r\dot{r}}{\tau\mathcal{H}(r,t)})\right)\right] \quad \text{and} \quad \dot{\phi} = \frac{\mathcal{L}_z}{r^2\sin^2\theta}, \tag{11}$$

with the following normalization condition for null/light-like geodesics:

$$g_{\mu\nu}\dot{x}^\mu\dot{x}^\nu = 0. \tag{12}$$

By using Equation (11) in Equation (12), and replacing $\theta = \frac{\pi}{2}$ and $\dot{r} = 0$ we obtain the following expression for effective potential $\mathcal{V}_E$:

$$\mathcal{E}^2 = (1 - \frac{2\gamma t}{\tau})\left(1 - \frac{2\mathcal{M}(1 - \frac{\gamma t}{\tau})}{r}\right)\frac{\mathcal{L}_z{}^2}{r^2} = \mathcal{V}_E \tag{13}$$

**Table 1.** First integrals.

| Generators | Conservation Laws |
|---|---|
| $\mathbf{Y}_1$ | $\mathcal{E} = (1 - \frac{\gamma t}{\tau})\left(1 - \frac{2\mathcal{M}(1 - \frac{\gamma t}{\tau})}{r}\right)\dot{t} + \frac{\gamma r\dot{r}}{\tau\left(1 - \frac{2\mathcal{M}(1 - \frac{\gamma t}{\tau})}{r}\right)} + \frac{\gamma s}{\tau}\mathcal{L}$ |
| $\mathbf{Y}_2$ | $\mathcal{L} = \left(1 - \frac{2\mathcal{M}(1 - \frac{\gamma t}{\tau})}{r}\right)\dot{t}^2 + \frac{\dot{r}^2}{\left(1 - \frac{2\mathcal{M}(1 - \frac{\gamma t}{\tau})}{r}\right)} + r^2(\dot{\theta}^2 - \sin^2\theta\dot{\phi}^2)$ |
| $\mathbf{Y}_3$ | $\mathcal{L}_z = r^2\sin^2\theta\dot{\phi}$ |
| $\mathbf{Y}_4$ | $\phi_4 = r^2(\dot{\theta}\cos\phi - \dot{\phi}\sin\theta\cos\theta\sin\phi)$ |
| $\mathbf{Y}_5$ | $\phi_5 = r^2(\dot{\theta}\sin\phi + \dot{\phi}\sin\theta\cos\theta\cos\phi)$ |

Figure 1 reports the profiles of effective potential against the different involved physical parameters. In Figure 1a, we noticed that the effect of the photons grows gradually for growing radial distances $r$. However, for increasing values of angular momentum $\mathcal{L}_z$, $\mathcal{V}_E$ shows dual nature. It is examined that for small domain of time coordinate, $\mathcal{V}_E$ and $\mathcal{L}_z$ behaves inversely. Contrarily, they are proportional to each other in large domain of the time coordinate. In Figure 1b, we observed that the higher influence of the time conformal factor $\gamma$ increases $\mathcal{V}_E$ for small values of time but they behave inversely for larger time coordinates.

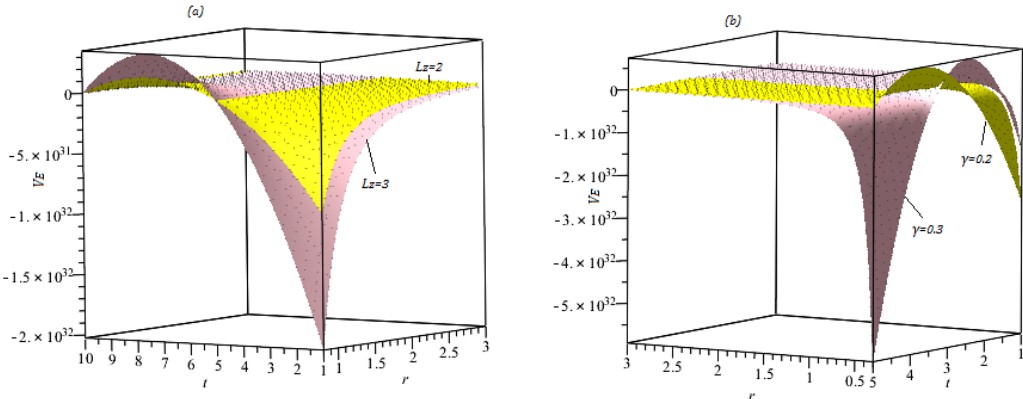

**Figure 1.** Profiles of effective potential of Equation (13) with (**a**). $\tau = 1$, $\gamma = 0.1$, $\mathcal{M} = 8M_{\odot}$ (**b**). $\tau = 1$, $\mathcal{L}_z = 1$, $\mathcal{M} = 8M_{\odot}$.

Expression for effective force can be calculated as:

$$\mathcal{F}_E = -\frac{1}{2}\frac{\partial \mathcal{V}_E}{\partial r}, \tag{14}$$

by using Equation (13) in (14) we obtained the following effective gravitational force for neutral motion of photons as:

$$\mathcal{F}_E = \frac{(1 - \frac{2\gamma t}{\tau})\mathcal{L}_z{}^2\left(r - 3\mathcal{M}(1 - \frac{\gamma t}{\tau})\right)}{r^4}. \tag{15}$$

It is shown in Figure 2 that the effective force is more attractive for large value of the time parameter $\gamma$ presented in Figure 2a. For constant value of angular momentum the effective force is directly proportional to the time parameter $\gamma$. Figure 2b shows the variation in the effective force corresponding to the variation in the angular momentum. The angular momentum is directly proportional to the effective force for small values of $r$ and $t$. However, for large value of $r$ and $t$ they are inversely proportional. Therefore, the effective force become weaken for large values of angular momentum for large $r$ and $t$. Additionally, we see from Figure 2b that the effective force is attractive in some region and repulsive in some other region.

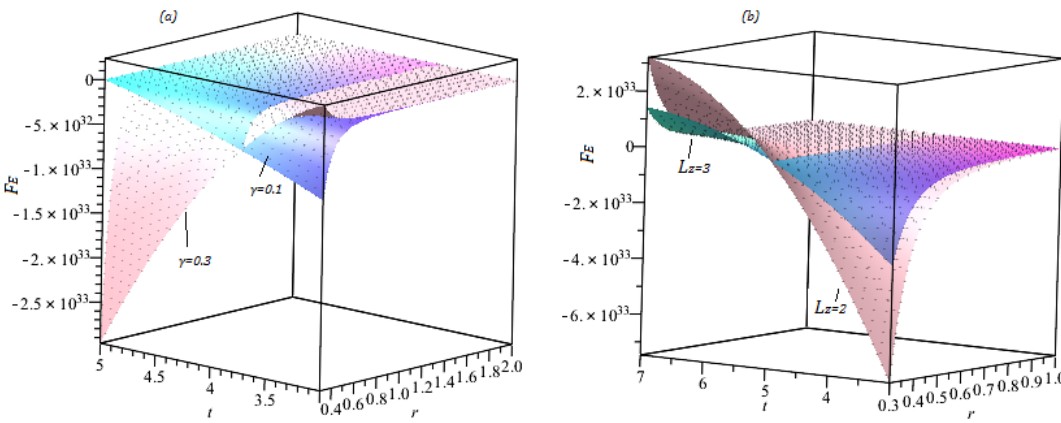

**Figure 2.** Profiles of effective force of Equation (15) (**a**). $\tau = 1$, $\mathcal{M} = 8M_{\odot}$, $\mathcal{L}_z = 1$ (**b**). $\tau = 1$, $\mathcal{L}_z = 1$, $\gamma = 0.1$, $\mathcal{M} = 8M_{\odot}$.

### 3.1. Stability of Circular Orbits

The convolution point of $\mathcal{V}_E$ lies in innermost circular orbit (ISCO). By taking $\mathcal{V}_{Er} = 0$ we obtain the (ISCO) for circular null geodesics as:

$$r_c = 3\mathcal{M}(1 - \frac{\gamma t}{\tau}).\tag{16}$$

To check the stability of orbits of evaporating Schwarzschild black hole we use Lyapunov exponent [28], i.e.,

$$\lambda = \left( \frac{-\frac{\partial^2 \mathcal{V}_E(r_c,t)}{\partial r_c^2}}{2\dot{t}^2(r_c)} \right)^{\frac{1}{2}} = \left( \frac{-\frac{\partial^2 \mathcal{V}_E(r_c,t)}{\partial r_c^2} r_c^2 \mathcal{H}(r,t)}{2\mathcal{L}_z^2} \right)^{\frac{1}{2}},\tag{17}$$

After some necessary calculations we will obtain the following form for Lyapunov exponent:

$$\lambda = \frac{(1 - \frac{\gamma t}{\tau})}{r_c^2} \sqrt{3\left( 6\mathcal{M}r_c(1 - \frac{\gamma t}{\tau}) - 8\mathcal{M}^2(1 - \frac{2\gamma t}{\tau}) - r_c^2 \right)}\tag{18}$$

In Figure 3a, we used numerical computations for Lyapunov exponent to determine the stability and instability of equatorial circular geodesics for null case. It is noted that inclusion of time factor decreases the stability of orbits.

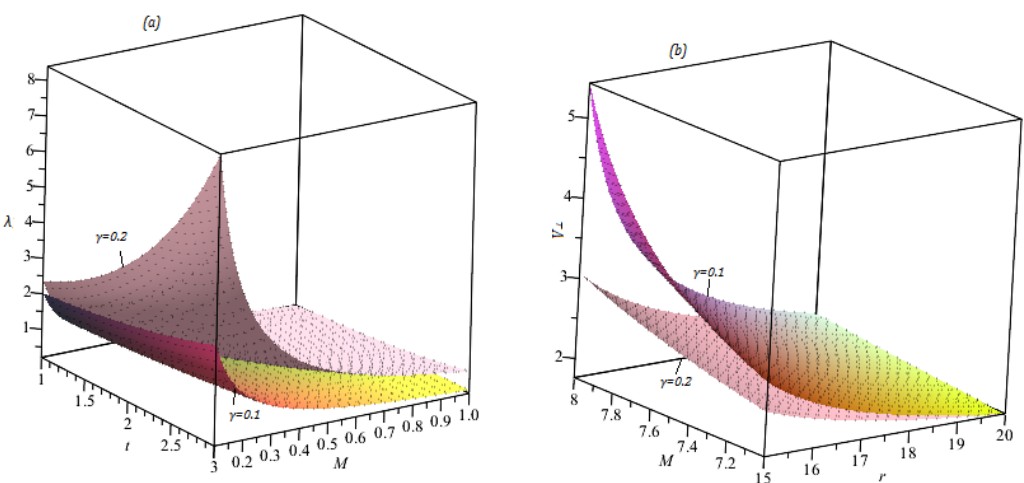

**Figure 3.** Profiles of the Lyapunov exponent and escape velocity of Equations (18) and (20) with (**a**). $\tau = 1$ (**b**). $\tau = 1$, $\mathcal{L}_z = 1$, $\gamma = 0.1$, $E = 10$.

### 3.2. Collision of Neutrally Moving Particles

In the case of collision, energy, and momentum of neutral photons will change. Here, we take angular momentum as conserved and energy as non-conserved quantity. After collision effective potential will take the form as:

$$\mathcal{V}_E = (1 - \frac{2\gamma t}{\tau})\left( 1 - \frac{2\mathcal{M}(1 - \frac{\gamma t}{\tau})}{r} \right)\left( \frac{\mathcal{L}_z + rv_\perp}{r} \right)^2.\tag{19}$$

The term $v_\perp$ is escape velocity of the photons. Expression for escape velocity can be written as:

$$v_\perp = -\frac{\mathcal{L}_z}{r} \pm \frac{r\mathcal{E}(1 + \frac{\gamma t}{\tau})}{\sqrt{\mathcal{H}(r,t)}}.\tag{20}$$

From Figure 3a, we see that the time parameter $\gamma$ decreases the stability of the orbits, because for larger values of $\gamma$, the Lyapunov exponent has large value while for small values of the $\gamma$ it has small values. The escape velocity decreases as we increase the time dependent parameter $\gamma$ which is shown in Figure 3b. Therefore, the time dependent parameter is inversely proportional to the escape velocity of the test particle. By using Equation (19) in (14), we obtain the following solutions for effective force:

$$\mathcal{F}_E = \frac{(1 - \frac{2\gamma t}{\tau})}{r^4}\left[ \mathcal{L}_z(\mathcal{L}_z + rv_\perp)\left(r - 2\mathcal{M}(1 - \frac{\gamma t}{\tau})\right) - H_o(r, t)\right] \tag{21}$$

where we have

$$H_o(r, t) = \mathcal{M}(1 - \frac{\gamma t}{\tau})(\mathcal{L}_z{}^2 + r^2 + r^2 v_e^2 + 2r\mathcal{L}_z v_\perp).$$

## 4. Null Geodesics for Charged Photons at Equatorial Plane

Let us assume the motion of charged photons effected by uniform agnatic field having strength $B$, which is present in the neighbourhood of evaporating Schwarzschild black hole. The Lagrangian in the present case is given as [29]:

$$\pounds = \frac{1}{2}g_{\mu\nu}u^\mu u^\nu + \frac{qA_\mu u^\mu}{m}. \tag{22}$$

In the above equation $m$ and $q$ refers to the mass of test particle and electric charge, respectively. For $\mathcal{B} = \frac{q\mathbb{B}}{m}$, we obtain the following constants of motion:

$$\dot{t} = \frac{(1 + \frac{\gamma t}{\tau})\mathcal{E}}{\mathcal{H}(r, t)} \qquad \text{and} \qquad \dot{\phi} = \frac{\mathcal{L}_z}{r^2} - \mathcal{B}. \tag{23}$$

Using Equation (23) in (12) we obtain the following expression for effective potential:

$$\mathcal{V}_E = (1 - \frac{2\gamma t}{\tau})\left(1 - \frac{2\mathcal{M}(1 - \frac{\gamma t}{\tau})}{r}\right)(\frac{\mathcal{L}_z}{r} - \mathcal{B}r)^2, \tag{24}$$

which leads to the following gravitational effective force:

$$\mathcal{F}_E = \frac{(1 - \frac{2\gamma t}{\tau})}{r^4}\left[ (\mathcal{L}_z{}^2 - \mathcal{B}^2 r^4)\left(r - 2\mathcal{M}(1 - \frac{\gamma t}{\tau})\right) - H_1(r, t)\right] \tag{25}$$

where

$$H_1(r, t) = \mathcal{M}(1 - \frac{\gamma t}{\tau})(\mathcal{B}^2 r^4 - 2\mathcal{L}_z\mathcal{B}r^2 + \mathcal{L}_z{}^2).$$

Figure 4 describes the profiles of effective force and effective potential $\mathcal{V}_E$ of charged photons. In Figure 4a, we observed that initially magnetic parameter $\mathcal{B}$ weaken the profiles of effective potential of charged photons but with the passage of time, $\mathcal{B}$ enhances $\mathcal{V}_E$. It is also detected that extending radial coordinates profiles of effective potential declines. In Figure 4b, we spotted that initially effective gravitational force and magnetic parameter strength are proportional but with the passage of time they shows inverse behavior. Moreover, stretching radial coordinates enhances the effects of effective gravitational force. After collision of charged colliding photons the effective potential will take the form as

$$\mathcal{V}_E = (1 - \frac{2\gamma t}{\tau})\left(1 - \frac{2\mathcal{M}(1 - \frac{\gamma t}{\tau})}{r}\right)(\frac{\mathcal{L}_z + rv_\perp}{r} - \mathcal{B}r)^2, \tag{26}$$

The effective force corresponding to Equation (27) is:

$$\mathcal{F}_E = \frac{(1 - \frac{2\gamma t}{\tau})}{r^4}\left[(\mathcal{L}_z + \mathcal{B}r^2)(\mathcal{L}_z - \mathcal{B}r^2 + rv_\perp)\left(r - 2\mathcal{M}(1 - \frac{\gamma t}{\tau})\right) - H_2(r,t)\right] \tag{27}$$

where

$$H_2(r,t) = \mathcal{M}(1 - \frac{\gamma t}{\tau})(\mathcal{L}_z + r^2 + rv_\perp - \mathcal{B}r^2)^2.$$

Expression for escape velocity after collision is calculated from Equation (26) as:

$$v_\perp = \mathcal{B}r - \frac{\mathcal{L}_z}{r} \pm \frac{r\mathcal{E}(1 + \frac{\gamma t}{\tau})}{\sqrt{\mathcal{H}(r,t)}}. \tag{28}$$

Figure 5 express the lines of escape velocity of colliding photons in different circumstances. Figure 5a reveals that $v_\perp$ is proportional to both magnetic strength $\mathcal{B}$ and radial coordinates $r$. However, in Figure 5b we observed that higher influences of angular momentum $\mathcal{L}_z$ decay the escape velocity of charge colliding photons.

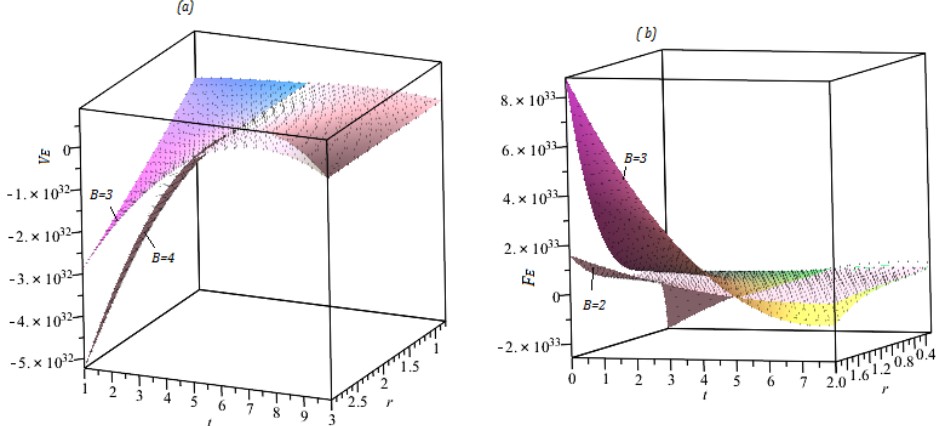

**Figure 4.** Profiles of effective potential and effective force of Equations (24) and (25) with (**a**). $\tau = 1$ b, $\tau = 1$, $\mathcal{L}_z = 1$, $\mathcal{M} = 8M_\odot$. (**b**) $\tau = 1$ b, $\tau = 1$, $\mathcal{L}_z = 1$, $\mathcal{M} = 8M_\odot$, $\gamma = 0.1$.

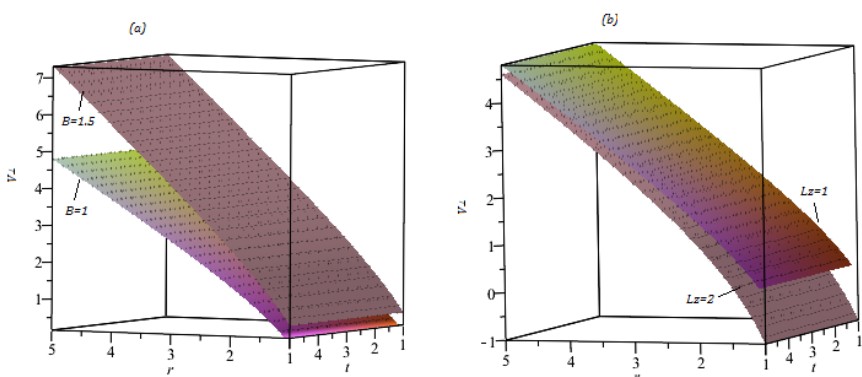

**Figure 5.** Profiles of escape velocity of Equation (28) with (**a**). $\tau = 1$ b. $t = 1$, $\mathcal{L}_z = 1$, $\mathcal{M} = 8M_\odot$, $\gamma = 0.1$ (**b**) $\tau = 1$, $t = 1$, $\mathcal{B} = 1$, $\mathcal{M} = 8M_\odot$, $\gamma = 0.1$.

## 5. Thermodynamics of Evaporating Schwarzschild Black Hole

The horizon of evaporating Schwarzschild black hole is calculated as:

$$r_h = 2\mathcal{M}(1 - \frac{\gamma t}{\tau}), \tag{29}$$

where the area of horizon of the black hole is:

$$\mathcal{A}_h = \int_o^{2\pi} \int_o^{\pi} (g_{\theta\theta} g_{\phi\phi})^{\frac{1}{2}} d\theta d\phi = 4\pi r_h{}^2 \tag{30}$$

which gives

$$\mathcal{A}_h = 16\pi \mathcal{M}^2 (1 - \frac{2\gamma t}{\tau}), \tag{31}$$

The Bekenstein–Hawking entropy [30] of black hole is:

$$\mathcal{S}_h = \frac{\mathcal{A}_h}{4}, \tag{32}$$

which is calculated as:

$$\mathcal{S}_h = 4\pi \mathcal{M}^2 \left(1 - \frac{2\gamma t}{\tau}\right). \tag{33}$$

Hawking temperature at the event horizon can be calculated by using the formula:

$$T_h = \frac{1}{4\pi} \frac{\partial \mathcal{H}(r,t)}{\partial r} |_{r=r_h}, \tag{34}$$

simplification of the above equation gives the following expression for Hawking temperature:

$$T_h = \frac{(1 + \frac{\gamma t}{\tau})}{8\pi \mathcal{M}}. \tag{35}$$

Komar energy [31] at event horizon of the evaporating Schwarzschild black hole which can be calculated as:

$$E_h = 2\mathcal{S}_h T_h, \tag{36}$$

for Schwarzschild evaporating black hole we obtain the following expression for Komar energy:

$$E_h = \mathcal{M}(1 - \frac{\gamma t}{\tau}). \tag{37}$$

Surface gravity [32] is the gravitational acceleration at the event horizon of a black hole which can be calculated as:

$$\mathcal{K}_h = \frac{1}{2} \frac{\partial \mathcal{H}(r,t)}{\partial r} |_{r=r_h}, \tag{38}$$

which can be written as:

$$\mathcal{K}_h = \frac{(1 + \frac{\gamma t}{\tau})}{4\mathcal{M}}. \tag{39}$$

Expression for pressure at the horizon of the black hole is:

$$p = \frac{\mathcal{K}_h}{8\pi}, \tag{40}$$

which can be written in following form:

$$p = \frac{(1 + \frac{\gamma t}{\tau})}{32\pi \mathcal{M}}, \tag{41}$$

From Figure 6 we can observe from (a) that profiles of entropy go down for increasing values of time and time-conformal parameter $\gamma$. On the one hand, in Figure 6b, it is noticed that temperature of horizon rise with the passage of time. Similarly higher influences of $\gamma$ increases the temperature of the horizon of the evaporating Schwarzschild black hole. Equations (35) and (39) relates temperature and surface gravity while Equations (33) and (37) give a relation between Komar energy and entropy of the horizon of evaporating Schwarzschild black hole as:

$$\mathcal{K}_h = 2\pi T_h \qquad \text{and} \qquad \mathcal{S}_h = 4\pi E_h{}^2,$$

which shows that whenever temperature grows at horizons the surface gravity of horizon grows more rapidly. On the other hand, entropy of the horizon decreases quadratically as the Komar energy decreasing function of time t.

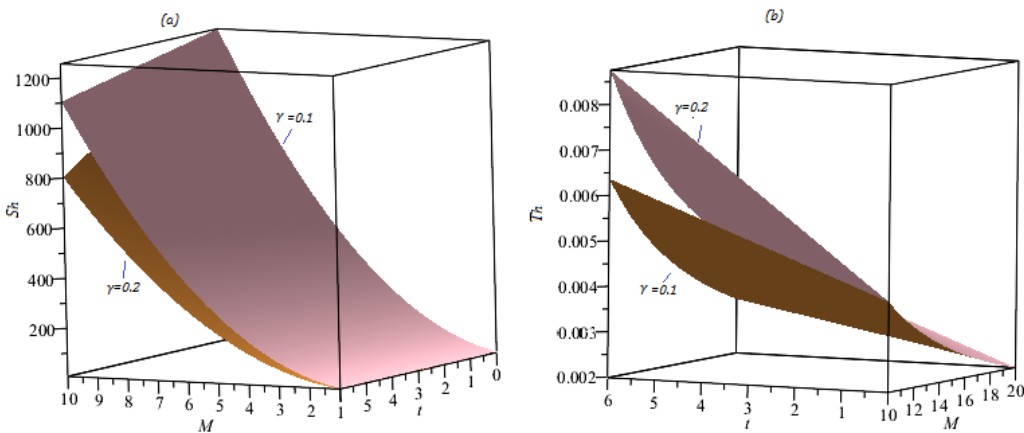

**Figure 6.** Profiles of (**a**) entropy at horizon with $\tau = 1$, $\mathcal{M} = 8M_\odot$, $\gamma = 0.1$, and (**b**) Hawking temperature with $\tau = 1$, $\mathcal{M} = 8M_\odot$, $\gamma = 0.1$.

## 6. Conclusions and Observations

In this manuscript, the geodesics motion for time-dependent Schwarzschild-like black hole are explored. Energy and momentum conservation laws are obtained by Noether symmetries. Profiles of effective potential, effective force and escape velocity are discussed for both neutral and charged particles. Stability of inner most circular orbits (ISCO) at equatorial plane is also discussed. Expressions for Hawking temperature, Bekenstein–Hawking entropy, Komar energy, surface gravity, and pressure at the perturbed horizon of evaporating Schwarzschild like black hole are calculated and discussed. We summarize the results as follows: the effective potential grows gradually along the radial distances. On the other hand, magnetic parameter and angular momentum both weaken the profiles of effective potential of charged particles initially, but with the passage of time they enhance the effective potential. Moreover, the time-dependent factor is proportional to the effective potential for a small time but for a large time it decreases the effective potential of the particle. The angular momentum also affects the effective potential. The effective force is more attractive for large value of the time parameter $\gamma$ presented in Figure 2a. For given value of angular momentum the effective force is directly proportional to the time parameter $\gamma$. The variation in the effective force corresponding to the variation in the angular momentum is given in Figure 2b. The angular momentum is directly proportional to the effective force for small values of $r$ and $t$. However, for large value of $r$ and $t$ they are inversely proportional. Therefore, the effective force become weak for large values of angular momentum for large $r$ and $t$. It is observed from Figure 2b that the effective force is attractive, as well as repulsive in different regions for different values of the angular momentum.

We observe that the time parameter $\gamma$ decreases the stability of the orbits, because for larger values of $\gamma$, the Lyapunov exponent has large values, while for small values of the

parameter $\gamma$ it has small values. The escape velocity is decreasing function of time $t$, as shown in Figure 3b. Therefore, the time dependent parameter is inversely proportional to the escape velocity of the test particle. Large value of time dependent term increases the stability of orbits. Komar energy for the perturbed Schwarzschild-like black, is a decreasing function of time $t$ as given in Equation (37). According to our investigation the surface gravity and pressure of the BH are increasing functions of time $t$ which are given in Equation (39) and Equation (41), respectively.

**Author Contributions:** M.A.K.: Data collection, calculation and writing, F.A.: Concepts, guidance and supervision, N.F.: Data analysis, interpretation and funding, M.A.E.-M.: Drafting, and critical revision of the article. All authors have read and agreed to the published version of the manuscript.

**Funding:** This research received no external funding.

**Data Availability Statement:** The data regarding the article is available from the authors on request.

**Acknowledgments:** The author Nahid Fatima would like to thank to the Prince Sultan University for their support and help.

**Conflicts of Interest:** There is no conflict of interest among the authors.

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
