# Peer review of "Particles Dynamics in Schwarzschild like Black Hole with Time Contracting Horizon"

_axioms, doi:10.3390/axioms12010034_

Round 1

Reviewer 1 Report (Previous Reviewer 3)

The paper verses on particle dynamics in Schwarzschild black hole. The authors calculate  the related quantities such as null geodesics, escape velocity, the Hawking temperature, the Bekenstein entropy and the Komar energy. I have some concerns on this paper that should be clarified to be reconsidered for publication such as the following raised points:

*The mass of the black hole in page 4 must be enumerated. The authors must clarify how the linear change in mass must influentiate Hawking evaporation, since any arbitrary time dependence may be introduced in mass. This is a capital statement that all the results of the paper depend. 

*Does the used metric satisfy Einstein equations?

* What is novel in this paper? 

*I call attention to the necessity of revision of some authors' phrases such as ''black holes absorbed negative energy'' or ''charged photon'' which does not make sense in this context. 

*The axes in Fig. (1) must be rescaled or the related quantities should be bold in typing for the sake of a clearer view from a reader perspective.

*The manuscript needs overall polishing in language, writing and scientific style. 

There are several typos throughout the text, e.g., equation() vs Eqn() vs. Eq.(), Figure() vs. Fig.() etc.

Author Response

Reviewer1: Comments and Suggestions for Authors

The paper verses on particle dynamics in Schwarzschild black hole. The authors calculate  the related quantities such as null geodesics, escape velocity, the Hawking temperature, the Bekenstein entropy and the Komar energy. I have some concerns on this paper that should be clarified to be reconsidered for publication such as the following raised points:

*The mass of the black hole in page 4 must be enumerated. The authors must clarify how the linear change in mass must influentiate Hawking evaporation, since any arbitrary time dependence may be introduced in mass. This is a capital statement that all the results of the paper depend.

Ans: The mass is enumerated now. The Black hole has five Noether symmetries given there in the paper. We perturbed the black hole spacetime metric with a general time dependent function and put the corresponding Lagrangian in the Noether symmetry equation. Fortunately, we get the linear function of time without losing any Noether symmetry of the black hole. If someone change this linear function of time, then it will lose at least one the Noether symmetry   which is related to the energy content in the given spacetime. So this linear function is introduced by a well-defined procedure to keep the symmetry structure of the black hole invariant.  

*Does the used metric satisfy Einstein equations?

Ans: Yes, the Metric of the black hole satisfy the Einstein field equations.

* What is novel in this paper? 

Ans: The novelty of the paper lies in the insertion of the time dependent function in the metric of the black hole without losing the symmetry structure. During the formation of gravitational waves, the black hole radiates its energy and momentum due which its mass and momentum are decreasing with passage of time. And the horizon is contracting with time because of the decrease in mass of the black hole.

*I call attention to the necessity of revision of some authors' phrases such as ''black holes absorbed negative energy'' or ''charged photon'' which does not make sense in this context.

Ans: We changed these phrases.

*The axes in Fig. (1) must be rescaled or the related quantities should be bold in typing for the sake of a clearer view from a reader perspective.

Ans: We did the necessary changes accordingly.

*The manuscript needs overall polishing in language, writing and scientific style. 

Ans: We went through the whole manuscript and improve its structure and language.

There are several typos throughout the text, e.g., equation() vs Eqn() vs. Eq.(), Figure() vs. Fig.() etc.

Ans: We removed such typos in the revised version of the paper.

Reviewer1: Comments and Suggestions for Authors

The paper verses on particle dynamics in Schwarzschild black hole. The authors calculate  the related quantities such as null geodesics, escape velocity, the Hawking temperature, the Bekenstein entropy and the Komar energy. I have some concerns on this paper that should be clarified to be reconsidered for publication such as the following raised points:

*The mass of the black hole in page 4 must be enumerated. The authors must clarify how the linear change in mass must influentiate Hawking evaporation, since any arbitrary time dependence may be introduced in mass. This is a capital statement that all the results of the paper depend.

Ans: The mass is enumerated now. The Black hole has five Noether symmetries given there in the paper. We perturbed the black hole spacetime metric with a general time dependent function and put the corresponding Lagrangian in the Noether symmetry equation. Fortunately, we get the linear function of time without losing any Noether symmetry of the black hole. If someone change this linear function of time, then it will lose at least one the Noether symmetry   which is related to the energy content in the given spacetime. So this linear function is introduced by a well-defined procedure to keep the symmetry structure of the black hole invariant.  

*Does the used metric satisfy Einstein equations?

Ans: Yes, the Metric of the black hole satisfy the Einstein field equations.

* What is novel in this paper? 

Ans: The novelty of the paper lies in the insertion of the time dependent function in the metric of the black hole without losing the symmetry structure. During the formation of gravitational waves, the black hole radiates its energy and momentum due which its mass and momentum are decreasing with passage of time. And the horizon is contracting with time because of the decrease in mass of the black hole.

*I call attention to the necessity of revision of some authors' phrases such as ''black holes absorbed negative energy'' or ''charged photon'' which does not make sense in this context.

Ans: We changed these phrases.

*The axes in Fig. (1) must be rescaled or the related quantities should be bold in typing for the sake of a clearer view from a reader perspective.

Ans: We did the necessary changes accordingly.

*The manuscript needs overall polishing in language, writing and scientific style. 

Ans: We went through the whole manuscript and improve its structure and language.

There are several typos throughout the text, e.g., equation() vs Eqn() vs. Eq.(), Figure() vs. Fig.() etc.

Ans: We removed such typos in the revised version of the paper.

Reviewer 2 Report (Previous Reviewer 4)

In this paper, the particles dynamics in a Schwarzschild like black hole with its time contracting horizon is argued. The discussions could be interesting and the investigations could be useful for related works. Thus, if the following points are reinvestigated very carefully, this work might be reconsidered for publication.

1. Many past related studies on black hole thermodynamics (including the works by the present authors). In comparison with these past works, the novel point and important results should be explained more explicitly. Namely, the differences of this paper from the past related works must be seen more clearly.   

2. A time dependent term is introduced to the horizon of black hole with the approximate Noether symmetry equation, so that the effective potential, effective force and all the dynamical features of the black hole can be affected. Do the astrophysical consequences depend on the way of introducing the time dependent term strongly? If it is the case, how can the genericity of the present analyses be verified?

3. In addition, null geodesics for neutral and charged particles are explored and the profiles of escape velocity of colliding particles are also investigated. Related to the point 2, from these analyses, what fundamental physics on gravitation can be found?

4. The stability of the orbits of the black hole is argued with the Lyapunov exponent. In addition, the Hawking temperature, Bekenstein entropy and Komar energy and specific energy at horizon of the black hole are studied. Such investigations would be a kind of an ordinary way in the study of black hole thermodynamics in the literature. What are the novel methods to consider the quantum aspects of black holes in comparison with the ordinary ones?

Author Response

Reviewer2:

Comments and Suggestions for Authors

In this paper, the particles dynamics in a Schwarzschild like black hole with its time contracting horizon is argued. The discussions could be interesting, and the investigations could be useful for related works. Thus, if the following points are reinvestigated very carefully, this work might be reconsidered for publication.

1. Many past related studies on black hole thermodynamics (including the works by the present authors). In comparison with these past works, the novel point and important results should be explained more explicitly. Namely, the differences of this paper from the past related works must be seen more clearly. 

Ans1:  The difference is explained in the introduction. In previous works we introduced a time conformal factor in the Schwarzschild spacetime without losing its symmetry structure. Here in this case, we only perturbed the horizon of the black hole a time dependent term without losing the symmetry structure of the black hole. In this case the horizon became time dependent and decrease linearly with the passage of time. Physically it is possible because when the black hole loses its energy and momentum in the form of gravitational waves its mass and energy decrease, and the horizon shrink.    

2. A time dependent term is introduced to the horizon of black hole with the approximate Noether symmetry equation, so that the effective potential, effective force and all the dynamical features of the black hole can be affected. Do the astrophysical consequences depend on the way of introducing the time dependent term strongly? If it is the case, how can the genericity of the present analyses be verified?

Ans2: As the black hole radiates it energy and momentum in the form of gravitational waves, therefore its mass and energy decreases with passage of time and the horizon shrinks toward the singularity. Which means that the horizon is a function of time ‘t’. And it is proved experimentally that gravitational waves exist and carry energy from the sources.

3. In addition, null geodesics for neutral and charged particles are explored and the profiles of escape velocity of colliding particles are also investigated. Related to the point 2, from these analyses, what fundamental physics on gravitation can be found?

Ans3: The Null geodesics, and escape velocity both become dependent on time. Using these geodesics the effective potential and effective force become dependent on time and the they are decreasing functions of time t.

4. The stability of the orbits of the black hole is argued with the Lyapunov exponent. In addition, the Hawking temperature, Bekenstein entropy and Komar energy and specific energy at horizon of the black hole are studied. Such investigations would be a kind of an ordinary way in the study of black hole thermodynamics in the literature. What are the novel methods to consider the quantum aspects of black holes in comparison with the ordinary ones?

Ans4:  The stability of the black hole decrease as it evaporates.

The Entropy and Hawking radiation are quantum effects, in our work both of them become the functions of time t.   

The Bekenstein-Hawking entropy is directly proportional to the surface area of the black hole. In our case the surface area of the black hole horizon is a decreasing function of time t, therefore the entropy should decrease with the passage of time because the mass of the black hole in our case is decreasing with time.

It is proved by Hawking that the temperature increasing with the evaporation of black hole. In our case the temperature is linearly increasing function of time t.

The Komar energy is also a decreasing function of time.  

By this investigation it is evident that the black hole stability, Bekenstein-Hawking entropy, Hawking temperature and Komar energy etc. become time dependent function and they are changing with passage of time

Round 2

Reviewer 1 Report (Previous Reviewer 3)

The author made the necessary efforts to improve the paper and they attented satisfactorily to all my raised points. The paper can be accepted for publication.

Reviewer 2 Report (Previous Reviewer 4)

The authors' answers to the review report are appreciated very much.
In the revised manuscript, the points suggested in the review report
have been reconsidered. Thus, this paper can be accepted for publication
in Axioms.

This manuscript is a resubmission of an earlier submission. The following is a list of the peer review reports and author responses from that submission.

Round 1

Reviewer 1 Report

The authors do not mention, and seem ignorant of, the Vaidya metric, which they partially duplicate. Their results of geodesics have been published long ago.

Unlike the Vaidya metric, they do not bother to demonstrate whether their metric satisfies Einstein's equation.

In addition, their linear mass change is not appropriate for Hawking evaporation, where the temperature changes.

Reviewer 2 Report

The paper studied Schwarzschild black holes with a time-dependent mass and computed various quantities for such black holes. My comments about this paper are as follows:
  1. The time dependence of the mass of the black hole is given in the equation between equations (2) and (3). However, it is not clear what the origin of this time dependence is, which clearly is not from Hawking radiation (since the time dependence in Hawking radiation is different). Maybe the authors can give some physical situations where such time dependence is possible. Another minor point is that the meaning of curly M in that equation should be clearly stated.
  2. It is not clear what is novel in this paper. One can introduce various time dependence into the black hole mass, and easily compute all the quantities studied in this paper. Maybe the authors can explain what phenomenon is unique to the time dependence studied in this paper. 
  3. Some statements in this paper seem confusing. For example, the authors said that “black holes absorbed negative energy”, but I think we usually just say black holes radiate. Another one is that the authors said “charged photon” in the paper, but clearly, in Electromagnetism, photons are not charged.
  4. There is a typo in equation (18).

Reviewer 3 Report

The paper has its merits but I believe that this work does not meet the scopus of the special issue named ''Computational mathematics and mathematics physics" focused on discrete and continuum mechanics and their developments. The submitted paper verses essentially on gravitational aspects of standard general relativity. I suggest that the authors may consider to submit their paper as a regular paper in Axiom journal that has a more general list of topics.

Reviewer 4 Report

In this paper, a particles dynamics is explored in a Schwarzschild like black
hole. In particular, the contracting horizon in time is studied. The discussions could be interesting and the discussions could be useful for related works. Thus, if the following points are re-investigated very carefully, this work might be reconsidered for publication.

1) There must exist a number of the past related works on dynamics around
the Schwarzschild black hole. By comparing with these past studies, the new points, important results and the novel method to find the observational constraints should be explained more explicitly, so that the originality of this article could be clarified. This is the most crucial point in this report.

2) It is stated that a time dependent term is introduced in the horizon of black hole without losing its symmetry structure by using the approximate Noether symmetry equation. What are the astrophysical positive reasons why such a configuration should be explored?  

3) It is mentioned that the Lyapunov exponent is used to check the stability of the orbits of the black hole. In addition, the Hawking temperature, the Bekenstein entropy and the Komar energy and specific energy at horizon of the black hole are examined. From these investigations, what fundamental physics of the strong gravity regime as well as black hole physics and thermodynamics can be found?

4) It is recommended that the presentations of the whole manuscript including Abstract and Conclusions should be rechecked. In particular, Conclusions should be describe by using ordinary sentences and not using only the items